# Anesthesia Management via an Automated Control System for Propofol, Remifentanil, and Rocuronium Compared to Management by Anesthesiologists: An Investigator-Initiated Study

**DOI:** 10.3390/jcm12206611

**Published:** 2023-10-19

**Authors:** Osamu Nagata, Yuka Matsuki, Shuko Matsuda, Keita Hazama, Saiko Fukunaga, Hideki Nakatsuka, Fumiyo Yasuma, Yasuhiro Maehara, Shoko Fujioka, Karin Tajima, Ichiro Kondo, Itaru Ginoza, Misuzu Hayashi, Manabu Kakinohana, Kenji Shigemi

**Affiliations:** 1Department of Anesthesiology and Reanimatology, University of Fukui, Fukui 910-1193, Japan; o-nagata@fa2.so-net.ne.jp (O.N.);; 2Department of Anesthesiology and Intensive Care Medicine, Hiroshima City Hiroshima Citizens Hospital, Hiroshima 730-8518, Japan; 3Department of Anesthesiology and Intensive Care Medicine, Kawasaki Medical School, Kurashiki 701-0192, Japan; 4Department of Anesthesiology, Center Hospital of the National Center for Global Health and Medicine, Tokyo 162-8655, Japan; 5Departments of Anesthesiology, The Jikei University School of Medicine, Tokyo 105-8461, Japan; 6Department of Anesthesiology, University of the Ryukyu, Okinawa 903-0213, Japan

**Keywords:** anesthesia, automated control system, closed-loop control, propofol, remifentanil, rocuronium

## Abstract

Background: We previously developed an automated total intravenous anesthesia control system that uses new closed-loop system algorithms to administer propofol, remifentanil, and rocuronium based on the bispectral index and train-of-four data. We recently improved this automated control system by adding a safety mechanism and using a modified monitoring device. Methods: Patients scheduled for elective surgery were randomly assigned to closed-loop feedback control (automatic group) or the manual administration of propofol, remifentanil, and rocuronium (manual group). The proportion of time during which the proper management of three-agent anesthesia was maintained during surgery was determined as the primary endpoint. Results: The proportion of time during which the three components of sedation, analgesia, and muscle relaxation were adequately controlled was 87.21 ± 12.79% in the automatic group, which was non-inferior to the proportion of 65.19 ± 20.16% in the manual group (*p* < 0.001). Adverse events during the operative or postoperative observation periods were significantly less frequent in the automatic group (54 patients, 90.0%) than in the manual group (60 patients, 100.0%; *p* = 0.027). Conclusion: Our three-agent automated control system, which features an improved muscle relaxation monitor and safety mechanism added to the basic control algorithms, maintained sedation, analgesia, and muscle relaxation appropriately in a manner non-inferior to anesthesiologists without compromising safety.

## 1. Introduction

Although the development of systems that mechanize the administration of intravenous anesthetics based on various biological information has been reported [1,2,3,4,5,6,7,8,9,10], there are no commercially available systems yet. We previously developed an automated total intravenous anesthesia control system that uses new closed-loop system algorithms to administer propofol, remifentanil, and rocuronium based on the bispectral index (BIS) and train-of-four (TOF) data [11,12,13,14]. Although this system was able to evaluate whether the control algorithm was working properly, it did not include safety mechanisms or other mechanisms required in actual clinical settings. Therefore, on the assumption that these systems would be used by many anesthesiologists if they entered the market, we developed an improved control software with several additional safety mechanisms. In this study, we examined the utility and safety of this new automated control system that achieves total intravenous anesthesia using three drugs.

## 2. Materials and Methods

### 2.1. Ethics Approval

All protocols were approved by the ethics committees of the University of Fukui Hospital and were conducted in accordance with the Declaration of Helsinki. Based on the results of the ethics review, the implementation of the study was approved by the other four hospitals in accordance with the Clinical Trials Act of Japan. The trial was registered with the Japan Registry of Clinical Trials (jRCT 2052190124). The authors affirm that the human research participants provided informed consent to participate in the study and for the publication of their data.

This was a multicenter, randomized, single-blinded, parallel group study conducted from March to September 2020 in patients scheduled for surgery with general anesthesia (i.e., abdominal, orthopedic, gynecological, and urological surgery). The inclusion criteria were:(1)Patients ≥20 years old at the time of providing informed consent;(2)Patients with an American Society of Anesthesiologists Physical Status (ASA-PS) 1–3.

The exclusion criteria were as follows:(1)A history of hypersensitivity to propofol, remifentanil, rocuronium, or sugammadex;(2)Patients scheduled to undergo surgery with hypothermia;(3)Patients scheduled to undergo cardiovascular surgery;(4)Patients scheduled to receive a nerve block or epidural anesthesia;(5)Pregnancy or breastfeeding.

The patients were randomly assigned to the automatic or manual group at a ratio of 1:1. A minimization method with the allocation factors of site, sex, and ASA-PS was used.

### 2.2. Anesthesia Procedures

No premedication was administered. After the patient was taken into the operating room, an electrocardiograph, non-invasive sphygmomanometer, pulse oximeter, and forehead BIS Quatro Sensor^®^ (Covidien Japan, Tokyo, Japan) were connected to the patient. Electromyograph electrodes connected to a muscle relaxation module (AF-200; Nihon Kohden, Tokyo, Japan) were attached to the forearm.

### 2.3. Manual Group

A remifentanil infusion was initiated at 0.5 µg·kg^−1^·min^−1^, based on an ideal body weight (IBW) derived from a body mass index (BMI) of 22 kg·m^−2^, at least 3 min after the initial administration of oxygen via a mask. About 4 min after the initial administration of remifentanil, a target-controlled infusion (TCI) of propofol was initiated with a plasma concentration of 3 μg·mL^−1^. After the patient fell asleep, rocuronium (0.6 mg·kg^−1^) was administered to achieve muscle relaxation, and then the patient’s trachea was intubated. The anesthesiologist adjusted the target concentration of propofol for a BIS value in the range of 35–55, adjusted the remifentanil infusion rate with reference to hemodynamic parameters (blood pressure and heart rate) during anesthesia (with an upper limit of 2 μg·kg^−1^·min^−1^), and adjusted the rocuronium infusion rate (initially 7 μg·kg^−1^·min^−1^) to maintain a train-of-four (TOF) count of 1 on muscle relaxation monitor data.

### 2.4. Automatic Group

The remifentanil infusion was started at 0.5 µg·kg^−1^·min^−1^, based on an IBW derived from a BMI of 22 kg·m^−2^. After about 5 min, propofol was administered at 0.5 mg·kg^−1^ for 10 s, with an infusion of propofol at 10 mg·kg^−1^·h^−1^ after the patient lost consciousness, according to package inserts. Once muscle relaxation was achieved with 0.6 mg·kg^−1^ of rocuronium, the automatic control of propofol, remifentanil, and rocuronium was started using the algorithms described below. These three agents, administered by the investigational device, were delivered according to their package inserts. An anesthesiologist continually monitored the patient to guard against over- or underdosing even during automatic administration. The control algorithms for propofol, remifentanil, and rocuronium were essentially those used in the previous study [14].

## 3. Propofol Control Algorithm

The estimated target effect-site concentration of propofol (esTEC-_P_) required to obtain the target BIS value can be calculated using the pharmacokinetic parameters of Marsh et al. [14] by collecting the paired data of the BIS and propofol concentration from the start of the propofol infusion (before the loss of consciousness) and fitting a sigmoid regression curve to the obtained paired data. The esTEC-_P_ that can achieve a BIS = 45 (esTEC-_P45_) was calculated in real time, and the continuous infusion rate of propofol was adjusted to achieve the esTEC-_P45_ in each patient. The software was modified to add the following tasks to handle urgent situations:Increase the target propofol concentration through the TCI over time when the BIS measurement precision is low (i.e., when electromyography exceeds the upper limit, the signal quality index is below the lower limit, or the BIS values are invalid);Temporarily increase the target propofol concentration when BIS values increase;Set a lower limit for the target propofol concentration (1.4 µg·mL^−1^ for patients <80 years old and 1.0 µg·mL^−1^ for patients ≥80 years old).

### 3.1. Remifentanil Control Algorithm

The paired data of the effect-site concentration of remifentanil, calculated using the pharmacokinetic parameters by Minto et al. [15] and the esTEC-_P45_, were collected from the start of remifentanil infusion, and an equipotential curve for analgesia (*x*) and sedation (*y*) was calculated via a regression analysis, using a rectangular hyperbolic function (*y* = *c*/(*x* − *a*) + *b*). Based on this isodynamic curve, even if the concentration of remifentanil is increased, the decreases in the esTEC-_P45_ would be less; that is, with respect to the remifentanil concentration at the point at which the absolute value of the slope becomes smaller, even if the analgesic effect is increased further, the decreases in the necessary dose of sedative in the individual patient would be slight. In other words, it suggests that the analgesic effect would level off. As such, this concentration is termed the “estimated maximal individual concentration (esMIC)”, since it is the estimated maximal concentration for achieving a realistic analgesic effect in an individual patient. Such a coordinate can be determined by assigning a percentage of the residual value (δ [%]) for the difference between the vertex and asymptote, in addition to the vertex (a + √c, b + √c) and asymptote (y = b) obtained from the regression curve (isodynamic curve) at a slope of −1. In the present study, the esMIC of remifentanil was calculated at a percentage of the residual value (δ) of 20% (esMIC_20_) [14].

The software was modified to add the following:The remifentanil infusion rate is changed to 0.5 µg·kg^−1^·min^−1^ at the start of surgical incision;A bolus of 50–100 μg of remifentanil is administered when the estimated maximal individual concentration (esMIC) (the details of the esMIC are described in our previous publication [14]) value increases suddenly;The propofol concentration is increased and the target remifentanil concentration is temporarily increased to 4.0 ng·mL^−1^ when the BIS is ≥70;When the esMIC decreases, the remifentanil concentration is lowered more gradually than in the previous study [14].

### 3.2. Rocuronium Control Algorithm

After the initial dose of rocuronium at 0.6 mg·kg^−1^, continuous rocuronium infusion was initiated to maintain the estimated blood concentration of rocuronium calculated using pharmacokinetic parameters by Wierda et al. [16] at the target level, which was defined as the effect-site rocuronium concentration at which the TOF count recovers to 1 (TOF1) on the muscle relaxation monitor. The rocuronium administration was controlled in the present study to achieve TOF1, whereas the administration of the muscle relaxant was controlled on the basis of %T1 in the previous study [13].

These control algorithms were installed in AsisTIVA^®^ (Nihon Kohden, Tokyo, Japan).

## 4. Evaluations

### 4.1. Evaluation of Sedation

The deviation time was defined as the time during which the anesthesiologist judged the sedation to be unsuitable in reference to movement, blood pressure, heart rate, and other aspects of the patient or the time during which the BIS deviated from the target range (35–55) while reliable BIS values were available (signal quality index ≥ 80).

### 4.2. Evaluation of Analgesia

As in the previous study [14], insufficient analgesia was defined as a rapid increase in systolic blood pressure and a heart rate of 20% over baseline (i.e., 1–5 min before the onset of hemodynamic change) except during the 5 min after vasoactive agent administration (Figure 1).

### 4.3. Evaluation of Muscle Relaxation

The deviation of the muscle relaxation level (TOF count) from 1 during surgery was defined as insufficient muscle relaxation. Events of insufficient muscle relaxation (movement) were also counted on a by-patient basis.

### 4.4. Evaluation of the Three Components

Missing data time: As in the previous study [14], the missing data time was defined as the time during which one of sedation, analgesia, or muscle relaxation was missing and no remaining items were outside the target range.

The time period during which sedation, analgesia, or muscle relaxation was not maintained within the target range for each patient was determined using the evaluation method described above and was subtracted from the overall operation time to obtain the adequate anesthesia time. Then, the ratio of adequate anesthesia time to the overall operation time was calculated as the adequate anesthesia time ratio (Figure 2).

### 4.5. Efficacy Outcome Measures

The primary outcome was the ratio of adequate anesthesia time to the period from the start of surgery to end of surgery under intravenous anesthesia with the three drugs (propofol, remifentanil, and rocuronium).

### 4.6. Safety Outcome Measures

All adverse events reported in patients from the start of surgery to 48 h after the end of surgery were evaluated, regardless of the causal relationship with the system.

### 4.7. Sample Size Calculation

In the previous study [14], which investigated the non-inferiority of automatic control in 28 patients to manual control in 28 patients, the mean (standard deviation) proportion of time during which sedation, analgesia, and muscle relaxation were adequately maintained relative to the surgery time was 73.2% (17.2%) in the automatic group and 59.9% (29.1%) in the manual group. Based on these values, the sample size was set at 60 patients per group for a total of 120 patients, assuming a dropout rate of 20% and a switchover from automatic control to manual control rate of 5%, to achieve power of at least 90% with a non-inferiority margin of 5% and using a two-sample *t*-test with a one-sided level of significance of 2.5%.

### 4.8. Statistical Analysis

The times during which analgesia, sedation, and muscle relaxation were adequately maintained relative to the surgery time (secondary outcomes 1–5) and the time during which all three components were adequately maintained (primary outcome) were compared using Student’s *t*-test with a one-sided level of significance of 2.5% to determine whether the automatic control was non-inferior to manual control. A non-inferiority margin between the manual and automatic control groups of 5% was used. This margin represents a clinically acceptable difference based on the clinical experience of anesthesiologists.

Two-sided *p*-values were calculated for secondary outcome measures of 6–8, using Student’s *t*-test. The results are shown as mean ± standard deviation values. For the incidences of adverse events (the safety outcome measure), two-sided *p* values were calculated using Fisher’s exact probability test.

## 5. Results

A total of 123 patients were enrolled in the study, with 63 assigned to the automatic group and 60 to the manual group, but 3 dropped out before their registration for surgery, leaving 120 patients (60 in the automatic group and 60 in the manual group). Since 1 patient in the automatic group was discontinued during surgery, the analysis population contained 119 patients (59 in the automatic group and 60 in the manual group) (Figure 3). The patients’ characteristics are shown in Table 1. Age, sex, BMI, and ASA-PS score did not differ significantly between the groups.

### 5.1. Anesthetic Management Provided by the Automatic Control System

The time courses of measurements (BIS, blood pressure, and heart rate) and drug levels (propofol, remifentanil, and rocuronium effect-site concentrations) during surgery are shown separately for the automatic and manual groups (Figure 4). The BIS, blood pressure, and heart rate varied more in the manual group than in the automatic group. 

### 5.2. Efficacy Outcomes

The proportion of adequate anesthesia time in the FAS, as the primary outcome, was 87.21 ± 12.79% in the automatic group and 65.19 ± 20.16% in the manual group; automatic control was non-inferior to manual control (*p* < 0.001). 

All five of the secondary outcomes with one or more components of the primary outcome (sedation, analgesia, and muscle relaxation) in the automatic group were non-inferior to those in the manual group (Table 2).

The proportion of time during which the BIS was maintained within the target range (35–55) relative to the surgery time in the automatic group was non-inferior to that in the manual group (*p* < 0.001).

There were no intergroup differences in time from the administration of the neuromuscular antagonist (sugammadex) to the recovery of the TOF ratio to ≥0.9 or the time from the end of the propofol infusion to awakening (Table 3).

No serious adverse events CTCAE (Common Terminology Criteria for Adverse Events) of Grade 3 or higher were observed in the safety evaluation.

## 6. Discussion

The purpose of this study was to evaluate the utility and safety of a new automated control system equipped with safety mechanisms in addition to the automated control function using a simple control algorithm. Some anesthesiologists may be concerned about losing their jobs if these automated control systems were to become practical. To avoid such anxiety, this study, like previous studies, examined the non-inferiority of these systems with respect to primary endpoints. Automated control achieved via these devices showed non-inferiority compared with anesthesiologists for all three effects of sedation, analgesia, and muscle relaxation. Furthermore, it can be interpreted that the automatic group is superior to the manual group for items for which the point estimate of the difference between the two groups and the 97.5% one-sided confidence interval are greater than 0.

The proportion of time during which all three elements of sedation, analgesia, and muscle relaxation were controlled within the target range, the proportion of time during which sedation was controlled within the target range, and the proportion of time during which muscle relaxation was controlled within the target range were all higher in the present study than in a previous study [14] (73.24 ± 17.24% vs. 87.21 ± 12.79%, 93.90 ± 7.25% vs. 96.39 ± 4.49%, 78.97 ± 17.27% vs. 90.84 ± 12.60%). Although a direct comparison cannot be made, the software with the additional safety mechanisms in the present study is thought to have improved the control accuracy compared with the previous study [14].

In a detailed investigation of the automated control of sedation, there was only one individual for whom the proportion of time with adequate control was less than 80%. In this individual, even though the propofol concentration was at the lower control limit (1.4 μg·mL^−1^), there were many times when the BIS level was below the lower limit (35) of the control target range and very few times at which the BIS level exceeded the upper limit (55) of the control target range. In this system, a safety mechanism has was to avoid the underestimation of the administration of propofol that could result in the patient awakening during surgery, and so it is thought that there were times at which propofol continued to be administered at the lower control limit even when the BIS level fell below the lower control limit. However, even in individuals in the automatic group for whom the proportion of time with adequate control was low, ≥70% were within the target range, and there was less body movement or fewer other adverse events due to an unexpected BIS elevation. Thus, the addition of this algorithm does not seem to have caused any major disadvantages in the control of sedation, and the safety mechanism for avoiding a sudden lack of sedation is thought to be practical.

In the control of analgesia, the proportion of time with adequate analgesia was nearly 100% in both the automatic group and manual group, and so the control was nearly perfect. The lack of pain monitors has prevented the automatic control of analgesics such as remifentanil. Although pain monitors have been developed [17,18,19,20], none are extensively used in clinical settings as of yet. Faced with this limitation, we decided to adjust the dosage according to the balance between remifentanil and propofol based on analgesia–sedation interactions. This allowed for the automatic control of the remifentanil dosage without an analgesic monitor. The analgesic control achieved via the algorithm we used is also thought to be able to reliably maintain stable general anesthesia with sufficient administration of the anesthetic. At the same time, it is also thought to leave sufficient room for decreases in the administration of the anesthetic.

Regarding the automated control of muscle relaxation, there were individuals for whom the proportion of time with adequate control was low. The most important reason for the low accuracy of muscle relaxation control is the influence of physical noise from the surgeon’s movements and electrical noise from the electric scalpel on the data obtained from the muscle relaxation monitor. BIS monitors are equipped with noise-removal algorithms, but TOF monitors evaluate acceleration/electromyography and are not equipped with algorithms to remove various types of noise. If this point could be improved, it would be possible to improve the accuracy of the regulation of the muscle-relaxation state in automatic control. At the same time, muscle relaxation is of lower importance than sedation and analgesia, and not all surgeries require that a state of stable muscle relaxation be continuously maintained. Therefore, rather than using the automated control function for sedation, analgesia, and muscle relaxation for all general anesthesia patients, it is thought that the automated control function for muscle relaxation should be used only in situations in which it is truly necessary. 

Since this system had no problems with respect to safety assessment indicators, it is thought that the system itself can be used safely in clinical settings. On the other hand, it needs to be fully understood that this system cannot perform anesthetic control in the place of an anesthesiologist and that it is not a device that can satisfy the preferences of the anesthesiologist, such as an emphasis on analgesia or sedation. In other words, this system should be used by anesthesiologists with specialized knowledge and skills, and when it is used, its actions should be continuously monitored. If a problem occurs, such as an unexpected lack of sedation and analgesia, due to the malfunction of the various monitoring devices or trouble in the infusion circuit, the anesthesiologist in charge needs to respond quickly to resolve the issue.

Although occasional reports of the development of automated control systems for intravenous anesthetics based on closed-loop control have been seen [1,2,3,4,5,6,7,8,9,10], none are widely used in clinical settings. The reasons for this are that society’s acceptance of the use of this kind of system and the establishment of systems for the safe use of these devices are difficult issues. In Japan, guidelines for the proper use of these systems were published in the spring of 2023 by the Japanese Society of Anesthesiologists [21], a public organization of anesthesiologists, ahead of the commercial launch of this product. These guidelines establish criteria for institutions and users. The use of this system in accordance with these criteria is promising for both improving the quality of general anesthesia management with total intravenous anesthesia and the elimination of disparities in the quality of anesthesia.

The system investigated in this study has several limitations. First, the system responds to sudden increases in invasive stimulation. Enabling a bolus injection of remifentanil shortened the time of response to sudden increases in invasive stimulation. Although no awakening during surgery or unanticipated problematic movement occurred in the study, room for further improvement remains. Second, surgeries maintained using regional anesthesia such as epidural anesthesia or a nerve block were excluded from the study. A combination with epidural anesthesia or a nerve block reduces the required dose of the three agents, and the system must first be verified to operate in a stable manner when such procedures are used. Furthermore, anesthesiologists can obtain information about changes in surgical stress through good communication with the surgeon, but changes in surgical stress cannot be predicted in advance using this system. However, the amount of analgesic administered is greater with the analgesic administration algorithm of this system than with an adjustment by an anesthesiologist, and it has an added safety mechanism to quickly increase the concentration of the analgesic when surgical stress increases. With this, it is thought that times of inadequate analgesic effect can be minimized even when surgical stress intensifies rapidly. The study itself also had limitations. For example, only two subjects each with an ASA PS score of 3 were in the automatic group and manual group, so care must be exercised in its use for patients with severe systemic disease. Furthermore, although it cannot be available for cases involving the exclusion criteria listed in the Materials and Methods section, the accuracy of automatic regulation decreases, especially in situations in which the reliability of an electroencephalogram (BIS monitor) or muscle relaxation monitor (TOF monitor) is low. For this reason, automatic control becomes insufficient in situations in which there are diseases or drugs that affect BIS and TOF monitors.

## 7. Conclusions

We automatically controlled propofol and rocuronium based on biological information obtained from electroencephalogram monitors and muscle-relaxation monitors. Furthermore, remifentanil was automatically controlled by utilizing the interaction between analgesia and sedation. A commercial system was developed by adding safety mechanisms to these algorithms. The system was non-inferior to manual control by anesthesiologists and also had an improved control accuracy (especially in maintaining adequate muscle relaxation) compared to previous systems.

## Figures and Tables

**Figure 1 jcm-12-06611-f001:**
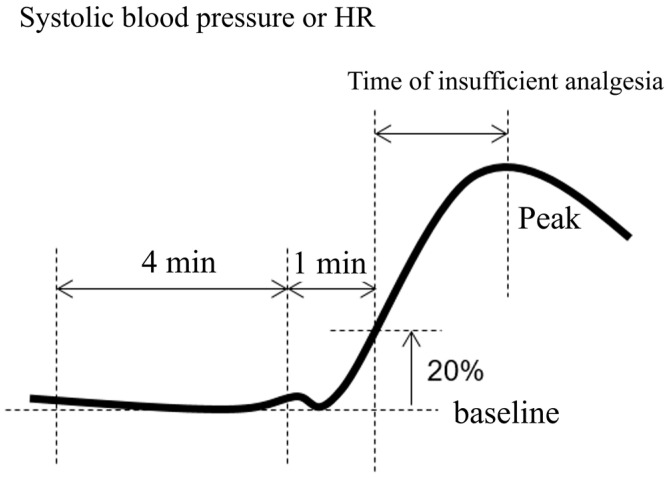
Definition of insufficient analgesia.

**Figure 2 jcm-12-06611-f002:**
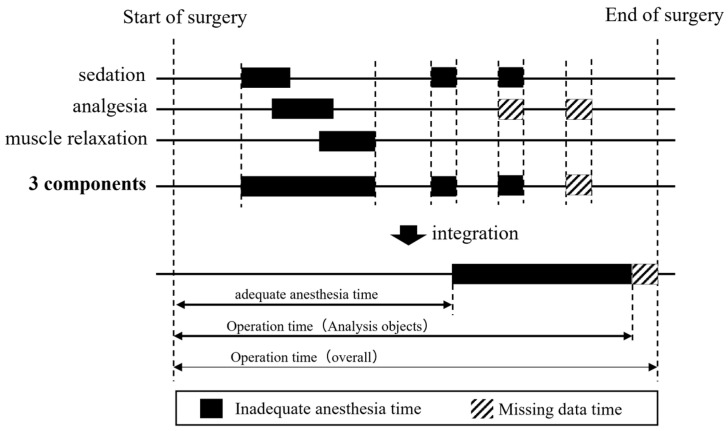
Definition of adequate anesthesia time.

**Figure 3 jcm-12-06611-f003:**
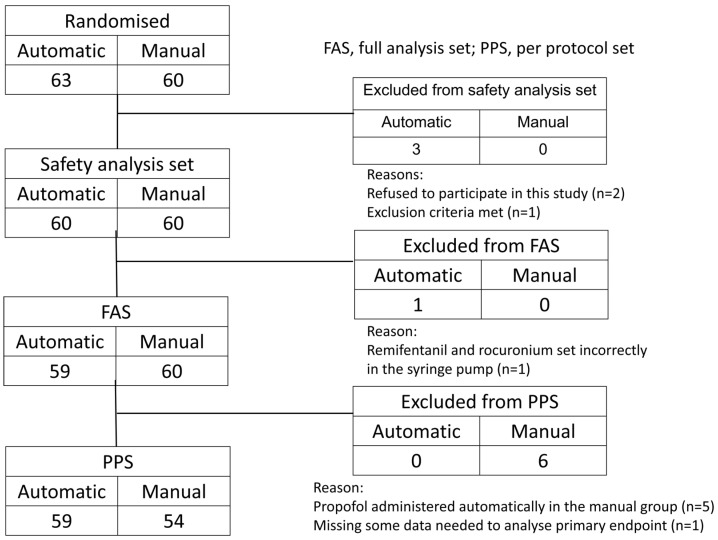
Flowchart of patient inclusion in the study.

**Figure 4 jcm-12-06611-f004:**
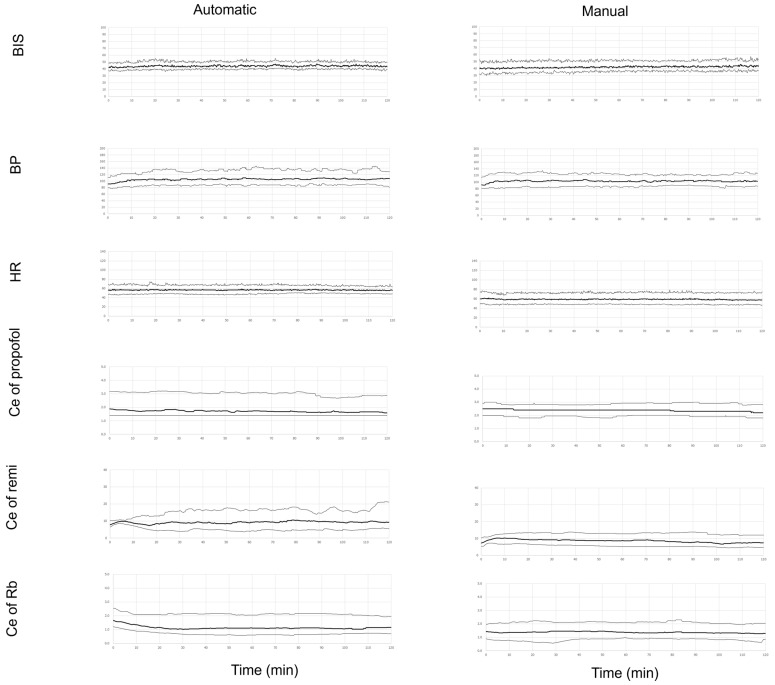
Bispectral index (BIS) values, systolic blood pressure (BP), heart rate (HR), and effect-site concentrations of propofol (Ce of propofol), remifentanil (Ce of remi), and rocuronium (Ce of Rb) for 2 h from the start of surgery. Data are given as median values with 10th and 90th percentiles.

**Table 1 jcm-12-06611-t001:** Patient characteristics.

Variable	Automatic (*n* = 60)	Manual (*n* = 60)	*p* Value
Sex			1.000
Male	18 (30.0%)	17 (28.3%)	
Female	42 (70.0%)	43 (71.7%)	
Age (year)	54.5 ± 14.9	55.7 ± 14.9	0.638
Height (cm)	160.52 ± 8.28	159.56 ± 9.13	0.548
Weight (kg)	60.37 ± 12.79	59.97 ± 11.92	0.860
BMI (kg·m^−2^)	23.29 ± 3.82	23.42 ± 3.58	0.843
ASA-PS			0.886
1	23 (38.3%)	20 (33.3%)	
2	35 (58.3%)	38 (63.3%)	
3	2 (3.3%)	2 (3.3%)	

ASA-PS, American Society of Anesthesiologists Physical Status; BMI, body mass index.

**Table 2 jcm-12-06611-t002:** Comparison of the primary outcome and its components (secondary outcomes 1–5).

	Automatic (*n* = 59)	Manual (*n* = 60)	Point Estimation	97.5% One-Tailed Confidence Interval	*p* Value
Ratio of time during which adequate control of all three factors was achieved (%)	87.21 ± 12.79	65.19 ± 20.16	22.03	15.87	<0.001
(1) Ratio of time during which adequate sedation control was achieved (%)	96.39 ± 4.49	89.94 ± 13.98	6.54	2.67	<0.001
(2) Ratio of time during which adequate analgesia control was achieved (%)	99.56 ± 1.86	99.93 ± 0.41	−0.37	−0.85	<0.001
(3) Ratio of time during which adequate muscle relaxation control was achieved (%)	90.84 ± 12.60	72.34 ± 20.03	18.51	12.42	<0.001
(4) Ratio of time during which adequate control of sedation and analgesia was achieved (%)	96.05 ± 4.73	89.87 ± 13.95	6.19	2.39	<0.001
(5) Ratio of time during which adequate control of sedation and muscle relaxation was achieved (%)	87.53 ± 12.81	65.26 ± 20.19	22.27	16.12	<0.001

**Table 3 jcm-12-06611-t003:** Comparison of other secondary outcomes 6–8.

	Automatic (*n* = 59)	Manual (*n* = 60)	*p* Value
(6) Ratio of time 35 ≤ BIS values ≤ 55 (%)	96.23 ± 4.93	88.08 ± 17.47	<0.001
(7) Time from muscle relaxant antagonist administration to recovery to TOF ratio > 0.9 (min)	2.51 ± 1.17	2.49 ± 1.16	0.909
(8) Time from the end of propofol infusion to awakening from anesthesia (min)	9.42 ± 3.64	8.40 ± 4.35	0.171

## Data Availability

The data that support the findings of this study are available from the corresponding author upon reasonable request.

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
