# Peer review of "Anesthesia Management via an Automated Control System for Propofol, Remifentanil, and Rocuronium Compared to Management by Anesthesiologists: An Investigator-Initiated Study"

_jcm, 2023, doi:10.3390/jcm12206611_

Round 1

Reviewer 1 Report

This study has an interesting topic about an automated closed-loop control system algorithm for the administration of propofol, remifentanil, and rocuronium. Overall, it was well written, and the automated closed-loop control system seemed superior to the existing manual method. 

It would be better to have a more specific explanation (or introduction) of this algorithm, such as the name of the system, program, interface, etc.

What is the difference between dividing secondary outcomes into 1-6 and 7 and 8 in the method and 1-5 and 6-8 in the table?

Is the difference between outcome 2 in Table 2 really p value <0.001? Just looking at the numbers, there doesn't seem to be much difference.

Line 252-254, as this was a non-inferiority study, it was described as non-inferior, but since it was actually superior, it seems better to describe it as superior.

In the safety evaluation (Table 4), incidence occurred in 90-100% of patients, which is likely due to the inclusion of postoperative pain. Reporting this itself is meaningless and it would be better to report only severe pain.

Author Response

Comments and Suggestions for Authors (Reviewer 1)

This study has an interesting topic about an automated closed-loop control system algorithm for the administration of propofol, remifentanil, and rocuronium. Overall, it was well written, and the automated closed-loop control system seemed superior to the existing manual method. 

→Thank you for your constructive comments.

It would be better to have a more specific explanation (or introduction) of this algorithm, such as the name of the system, program, interface, etc.

→In this system, sedative and analgesic administration is regulated using estimated Target Effect-Site Concentration (esTEC) for propofol, and analgesic administration is regulated using estimated Maximal Individual Concentration (esMIC). These control algorithms are installed in AsisTIVA® (Nihon Kohden, Tokyo, Japan). (P4L160)

What is the difference between dividing secondary outcomes into 1-6 and 7 and 8 in the method and 1-5 and 6-8 in the table?

→I apologize for the error in the description of the method; it has been corrected. (P5L213,P6L219)

Is the difference between outcome 2 in Table 2 really p value <0.001? Just looking at the numbers, there doesn't seem to be much difference.

→The p-values in this table are not the results of a simple two-group comparison (t-test), but the results of a noninferiority test under the conditions described in the methods.In order to convey the status of the comparison more accurately between the two groups, two columns showing the difference between the automatic group and the manual group and the lower limit of the one-sided 97.5% confidence interval have been added to Table 2. (P8L258- 259)

Line 252-254, as this was a non-inferiority study, it was described as non-inferior,

→In research that has set up a non-inferiority test in advance, it is contrary to research ethics to change the test method if significance is found after data analysis. On the other hand, we believe that it is necessary to make efforts to accurately convey information obtained from the data to the readers. Therefore, in Table 2 of the results, we have added the point estimate of the difference between the two groups and the 97.5% one-sided confidence interval. In the Discussion, we added that the automatic group can be interpreted as showing results superior to those of the manual group in items where these values are greater than 0. (P8 L274-277)

In the safety evaluation (Table 4), incidence occurred in 90-100% of patients, which is likely due to the inclusion of postoperative pain. Reporting this itself is meaningless and it would be better to report only severe pain.

→As you pointed out, events occurred in 90-100% of patients on safety evaluation (Table 4). Therefore, we have changed the statement to state that no serious adverse events (CTCAE (Common Terminology Criteria for Adverse Events) Grade 3 or higher) were observed. (P8L264-265)

Reviewer 2 Report

Thtopic of your study could be interesting enough for current medical literature. However, I am wondering about how you could relate different kind of surgery with a common opioids, sedatives and myorelaxants need. I can’t se an homogeneous patient’s population. What kind of surgery they underwent? Are those surgery similar in terms of pain sensation and kind of anesthesiology? There is a difference among different levels of consciousness you have toabolish if you decide to administer a general anesthesiology or a light/mild/deep sedation. In these latter circumstances you don’t even need a myorelaxant drug. So, my first question is about the variability of patients you have studied. I think you’d better border it rather than relate things without specific criteria.

Let’s analyze the introduction now. You chose to start it with this sentence: Advances in information technology have led to the successful application of automatic control in various fields, and several studies aiming at the practical application of automatic administration control systems for intravenous anesthetics in general anesthesia administered by anesthesiologists have been reported.” I don’t think this is the right choice.  What are these innovative ideas you are talking about? What is their field? Information or technology? It’s not clear what you are saying, and it is not even clear what message you want to highlight. You’d better introduce immediately the real topicof the study,and you could also spend some words about TCI, BIS and TOF, instead of making only a simple citation about what are they used for.

Let’s talk about the section “material and methods”. The subtitles and categories are adequate. However, in the sub-section “Evaluation of three components” you wrote: The times when sedation, analgesia, and muscle relaxation were not maintained 186 within the target ranges according to the evaluation methods listed above were deter- 187 mined on a by-patient basis. The duration from start to end of surgery minus the time 188 during which the sedative, analgesic, or muscle relaxant effect deviated was defined as 189 the time of adequate anesthesia. The proportion of this time relative to the operation pe- 190 riod (proportion of adequate anesthesia) was then calculated. (The detail of the evaluation 191 of the three components is described in our previous publication [14].)  Are you sure that colleagues from the whole world have read your previous work? I have not, actually. You must specify all the methods of each kind of evaluation you made and present them to the reader who should not be in need of looking about what you are writing about on his/her own. That last phrase (I put it in bold) you wrote is completely out of line.  You must erase it and make a clear presentation of those kind of evaluation criteria.You can’t make readers in need of changing paper so as to understand what evaluation criteria you chose to adopt.

However, the organization of the paper through these different sections and their sequence is right, even though you’d better put tables and graphs at the end of the full paper following the current guidelines about how to organize a scientific paper. Maybe an adjunctive division between tables vs graphs is more suitable than the sequence you made by mixing these two different types of figures. Anyway, all these tables are well constructed, even the ones related to the results, and are suitable to give an immediate message to any reader.  

The subdivision of each section through subtitles makes the paper accurate, but the discussion, even though well presented, needs you to check some aspects. Usually, discussions are thought to be short and made of a comparison between the results of one paper and the ones obtained by current medical literature thanks to other studies that have wondered about the same topic. You were definitely not able in doing this. You have to make it shorter so as to give a quick message to the reader. Then you have to make the comparison between your results and other similar results that have been presented in other studies, such us your previous one and note n.19 of your bibliography and so on. You wrote about a possible comparison between this current paper and the one you made previouslyHowever, is too hasty. It should be specified and discussed better, whereas you wrote about it as a not so interesting citation. Moreover, you did not clarify this comparison as an aim of this current study, so any reader could wonder why you wrote about it if this was not among your purposes. It is about results, aims, or what? What analogies are you searching for? At first you have to clarify the aims, results and nature of your previous work and then you can relate them to this current studyIt seems like an attempt you made to compensatea lack of data you got from this study trying to make the discussion better argued. But it does not work enough.

In addition, you don’t need to repeat the results you obtained as numbers and percentages, but you have to make a scientific comment on them heightening their meaning.

Moreoverall the limitations you found during this work, have to be listed at the beginning of thediscussion. 

Let’s discuss about your conclusion. Well, it is not assertive enough and it is even too short. Conclusions are usually needed to give an accurate answer to the aims of the studies, that is why you should highlight the results you got and make your conclusions argued in a better and more specific way.

Least but not last, you should check the grammar as there are some mistakes in each section of the text. For example, line 14 of the introduction; point 3 “exclusion criteria were as follows”; first line of the section called “anesthesia procedures”of material and methods and so on. There is a clear problem with the syntaxis, that is why you’d better check for English grammar rules. Moreover, you’d better use short phrases that can make readers able to quickly understand the real message of this work. 

Author Response

Comments and Suggestions for Authors (Reviewer 2)

The topic of your study could be interesting enough for current medical literature. However, I am wondering about how you could relate different kind of surgery with a common opioid, sedatives and myorelaxants need. I can’t see a homogeneous patient’s population. What kind of surgery they underwent? Are those surgery similar in terms of pain sensation and kind of anesthesiology? There is a difference among different levels of consciousness you have to abolish if you decide to administer a general anesthesiology or a light/mild/deep sedation. In these latter circumstances you don’t even need a myorelaxant drug. So, my first question is about the variability of patients you have studied. I think you’d better border it rather than relate things without specific criteria.

→Patients in this study were scheduled for surgery under general anesthesia (i.e., abdominal, orthopedic, gynecological, and urological surgery). In the anesthesia management of such patients, this system handles not only surgical invasiveness and individual differences (inter-individual variation), but also variations in surgical invasiveness and patient condition within the same patient (intra-individual variation).

Let’s analyze the introduction now. You chose to start it with this sentence: “Advances in information technology have led to the successful application of automatic control in various fields, and several studies aiming at the practical application of automatic administration control systems for intravenous anesthetics in general anesthesia administered by anesthesiologists have been reported.” I don’t think this is the right choice.  What are these innovative ideas you are talking about? What is their field? Information or technology? It’s not clear what you are saying, and it is not even clear what message you want to highlight. You’d better introduce immediately the real topics of the study, and you could also spend some words about TCI, BIS and TOF, instead of making only a simple citation about what are they used for.

→As suggested, I have simplified the introduction. (P1L41-P2L52)

TCI (target-controlled infusion) is described (P2L88), BIS is described (P2L81), and TOF is described (P2L95). Since TCI, BIS, and TOF are commonly used in routine general anesthesia, we have omitted a detailed explanation.

Let’s talk about the section “material and methods”. The subtitles and categories are adequate. However, in the sub-section “Evaluation of three components” you wrote: “The times when sedation, analgesia, and muscle relaxation were not maintained 186 within the target ranges according to the evaluation methods listed above were deter- 187 mined on a by-patient basis. The duration from start to end of surgery minus the time 188 during which the sedative, analgesic, or muscle relaxant effect deviated was defined as 189 the time of adequate anesthesia. The proportion of this time relative to the operation period 190 (proportion of adequate anesthesia) was then calculated. (The detail of the evaluation 191 of the three components is described in our previous publication [14].)”  Are you sure that colleagues from the whole world have read your previous work? I have not, actually. You must specify all the methods of each kind of evaluation you made and present them to the reader who should not be in need of looking about what you are writing about on his/her own. That last phrase (I put it in bold) you wrote is completely out of line.  You must erase it and make a clear presentation of those kind of evaluation criteria. You can’t make readers in need of changing paper so as to understand what evaluation criteria you chose to adopt.

However, the organization of the paper through these different sections and their sequence is right, even though you’d better put tables and graphs at the end of the full paper following the current guidelines about how to organize a scientific paper. Maybe an adjunctive division between tables vs graphs is more suitable than the sequence you made by mixing these two different types of figures. Anyway, all these tables are well constructed, even the ones related to the results, and are suitable to give an immediate message to any reader.  

→We have described the evaluation of three components. (P4L184-P5191) We inserted graphs (Figures 1,2) and explained the evaluation of 3 components. We also deleted the sentence, “The detail of the evaluation of the three components is described in our previous publication [14].”

Thank you for your advice.

The subdivision of each section through subtitles makes the paper accurate, but the discussion, even though well presented, needs you to check some aspects. Usually, discussions are thought to be short and made of a comparison between the results of one paper and the ones obtained by current medical literature thanks to other studies that have wondered about the same topic. You were definitely not able in doing this. You have to make it shorter so as to give a quick message to the reader. Then you have to make the comparison between your results and other similar results that have been presented in other studies, such us your previous one and note n.19 of your bibliography and so on. You wrote about a possible comparison between this current paper and the one you made previously. However, is too hasty. It should be specified and discussed better, whereas you wrote about it as a not so interesting citation. Moreover, you did not clarify this comparison as an aim of this current study, so any reader could wonder why you wrote about it if this was not among your purposes. It is about results, aims, or what? What analogies are you searching for? At first you have to clarify the aims, results and nature of your previous work and then you can relate them to this current study. It seems like an attempt you made to compensate a lack of data you got from this study trying to make the discussion better argued. But it does not work enough.

In addition, you don’t need to repeat the results you obtained as numbers and percentages, but you have to make a scientific comment on them heightening their meaning.

Moreover, all the limitations you found during this work, have to be listed at the beginning of the discussion. 

→Thank you for your detailed guidance on revising the manuscript.

Our paper demonstrated that remifentanil could be administered automatically despite the lack of pain monitoring. No one has successfully administered remifentanil automatically before. Therefore, our study cannot be compared to other studies. I have added an explanation of this point in the Discussion. (P9L303-308)

Let’s discuss about your conclusion. Well, it is not assertive enough and it is even too short. Conclusions are usually needed to give an accurate answer to the aims of the studies, that is why you should highlight the results you got and make your conclusions argued in a better and more specific way.

→We rewrote the conclusion based on your comments. (P10L363-369)

Comments on the Quality of English Language

Least but not last, you should check the grammar as there are some mistakes in each section of the text. For example, line 14 of the introduction; point 3 “exclusion criteria were as follows”; first line of the section called “anesthesia procedures” of material and methods and so on. There is a clear problem with the syntaxis, that is why you’d better check for English grammar rules. Moreover, you’d better use short phrases that can make readers able to quickly understand the real message of this work. 

→Thank you for pointing out the many points that non-English speaking researchers should keep in mind when writing papers in English. This manuscript was corrected by a professional, native English-speaking medical editor. Thank you for your advice.

Round 2

Reviewer 2 Report

This new revised paper is more accurate and better organized than the previous one.

You have been able to clarify the innovative system you want to talk about, that was not specified in the previous introduction. Moreover the aim of your study can be quickly understand as you highlighted it with simple and short sentences at the end of a well constructed introduction. 

This time the introduction is short enough and offer a right comparison between the results of your paper and the ones obtained by other studies that have wondered about the same topic

Let’s talk about the section “material and methods” as in the previous paper you wrote about a work you did relating it to this current study. (It was in the sub-section “Evaluation of three components”: “The detail of the evaluation 191 of the three components is described in our previous publication [14].)” 

In this current version you finally specified all the methods of each evaluation you made relating those evaluations with the results you have reported in your previous paper. Moreover, you were able to clarify the parameters throughout you established the level of sedation, insufficient anesthesia and missing data time. Their description related to specific ghraps (figure 1 and 2 on pages 4 and 5) make you able to give an immediate message to the reader about what is the true aim of your entire work. The figures are well done and offer a quick and fast idea about your work. 

In this current discussion you were able not to repeat the results you obtained as numbers and percentages, but you finally made the scientific comment on them and their meaning, that you failed in the previous version. However, you still failed in listing the study’ limitations at the beginning of your discussion.

 Let’s discuss about your conclusion. This time it answers to the aims of the study, however you should specified the last two sentences about muscle relaxation. I mean, why the authomatic system was not inferior to the other “especially in terms of muscle relaxation”? You should not put this result in brakets, but you should give a better argument about it.

Least but not last, you have checked the grammar following the right rules of English syntaxis and grammar as there are no mistakes and everyone can read and understand quickly what you are writing about without misundertanding the meaning due to long and wrong phrases.

In conclusion, you have to highlitght the limitations of the study at the begunning of your discussion and you have to give a better argument of your conclusion in term of muscle relaxation.

Author Response

This new revised paper is more accurate and better organized than the previous one.

You have been able to clarify the innovative system you want to talk about, that was not specified in the previous introduction. Moreover the aim of your study can be quickly understand as you highlighted it with simple and short sentences at the end of a well constructed introduction. 

This time the introduction is short enough and offer a right comparison between the results of your paper and the ones obtained by other studies that have wondered about the same topic

Let’s talk about the section “material and methods” as in the previous paper you wrote about a work you did relating it to this current study. (It was in the sub-section “Evaluation of three components”:

In this current version you finally specified all the methods of each evaluation you made relating those evaluations with the results you have reported in your previous paper. Moreover, you were able to clarify the parameters throughout you established the level of sedation, insufficient anesthesia and missing data time. Their description related to specific ghraps (figure 1 and 2 on pages 4 and 5) make you able to give an immediate message to the reader about what is the true aim of your entire work. The figures are well done and offer a quick and fast idea about your work. 

In this current discussion you were able not to repeat the results you obtained as numbers and percentages, but you finally made the scientific comment on them and their meaning, that you failed in the previous version. However, you still failed in listing the study’ limitations at the beginning of your discussion.

→I am happy for your understanding. Thank you for your constructive comments.

 Let’s discuss about your conclusion. This time it answers to the aims of the study, however you should specified the last two sentences about muscle relaxation. I mean, why the authomatic system was not inferior to the other “especially in terms of muscle relaxation”? You should not put this result in brakets, but you should give a

Least but not last, you have checked the grammar following the right rules of English syntaxis and grammar as there are no mistakes and everyone can read and understand quickly what you are writing about without misundertanding the meaning due to long and wrong phrases.

In conclusion, you have to highlitght the limitations of the study at the begunning of your discussion and you have to give a better argument of your conclusion in term of muscle

→We rewrote limitation of this study based on your comments. (P9L312-319,P10L362-367)